# Limited Benefit of Marine Protein Hydrolysate on Physical Function and Strength in Older Adults: A Randomized Controlled Trial

**DOI:** 10.3390/md19020062

**Published:** 2021-01-27

**Authors:** Linda Kornstad Nygård, Ingunn Mundal, Lisbeth Dahl, Jūratė Šaltytė Benth, Anne Marie Mork Rokstad

**Affiliations:** 1Faculty of Health Sciences and Social Care, Molde University College, P.O. Box 2110, 6402 Molde, Norway; ingunn.p.mundal@himolde.no (I.M.); anne.m.m.rokstad@himolde.no (A.M.M.R.); 2Department of Mental Health, Faculty of Medicine and Health Sciences, Norwegian University of Science and Technology (NTNU), P.O. Box 8905, 7491 Trondheim, Norway; 3Institute of Marine Research (IMR), P.O. Box 1870 Nordnes, 5817 Bergen, Norway; Lisbeth.Dahl@hi.no; 4Institute of Clinical Medicine, Campus Ahus, University of Oslo, P.O. Box 1171, Blindern, 0318 Oslo, Norway; jurate.saltyte-benth@medisin.uio.no; 5Health Services Research Unit, Akershus University Hospital, P.O. Box 1000, 1478 Lørenskog, Norway; 6Norwegian National Advisory Unit on Ageing and Health, Vestfold Hospital Trust, P.O. Box 2136, 3103 Tønsberg, Norway

**Keywords:** hydrolysate, fish protein, ageing, physical function, dietary assessment, seafood intake, healthy ageing

## Abstract

Age-related muscle wasting can compromise functional abilities of the elderly. Protein intake stimulates muscle protein synthesis; however, ageing muscle is more resistant to stimuli. This double-blinded, randomized, controlled trial is one of the first registered studies to evaluate the effects of a supplement of marine protein hydrolysate (MPH) on measures of physical function and strength. Eighty-six older adults received nutritional supplements containing 3 g of MPH or a placebo for up to 12 months. Short Physical Performance Battery (SPPB), grip strength and gait speed were measured, and dietary intake was registered at baseline, 6 months, and 12 months. No difference was found between the intervention and control groups in mean change in SPPB (independent sample *t*-test, *p* = 0.41) or regarding time trend in SPPB, grip strength, or gait speed (linear mixed model). The participants in our study were well functioning, causing a ceiling effect in SPPB. Further, they had sufficient protein intake and were physically active. Differences in physical function between those completing the intervention and the dropouts might also have created bias in the results. We recommend that future studies of MPH be carried out on a more frail or malnourished population.

## 1. Introduction

In 2019, 65-year-old Norwegian men could expect to live five years longer compared with 30 years ago [1]. With increasing lifespans, healthy ageing is crucial to maintain independence and reduce future healthcare costs for society. Maintenance of muscle strength and function plays an important role in healthy ageing [2]. Age-related muscle wasting (i.e., sarcopenia) is accompanied by loss of strength and can compromise the functional abilities and activity levels of the elderly [3,4].

One of the main mechanisms of muscle wasting is a reduction in muscle protein synthesis (MPS), and it appears that the muscles of older adults are more resistant to anabolic stimuli than the muscles of younger people. This implies that older muscles might need larger amounts of protein than younger ones to adequately stimulate MPS [2,3]. However, high protein intake might be challenging as ageing is often accompanied by decreased energy intake and loss of appetite, i.e., the anorexia of ageing [5]. 

Protein supplements have been shown to elicit gains in muscle strength in older people when used in combination with strength exercise [6]. However, the majority of muscle wasting happens in periods of low physical activity [7]. Thus, it is also interesting to study nutritional interventions alone, i.e., without combining them with exercise. Several signaling pathways are described where proteins, peptides, and amino acids stimulate MPS, and among the single amino acids, leucine and other branch-chained amino acids (BCAA) are especially active parts of these pathways [8]. Lean fish is a source of protein of good quality and tends to have a moderate to high content of BCAA [9]. Fish protein supplements in small doses have shown a positive effect on body composition in favor of muscle vs. fat in overweight people, both in a study using 3 g fish protein [10] and 1.4–2.8 g fish protein hydrolysate [11]. These doses are small, and the effect might be related to bioactive peptides, rather than the known effect of protein as a source of amino acids [12].

Small dose supplements could constitute a more feasible way of supplementing elderly and/or frail persons as the amount of intake is smaller and contributes less to the feeling of satiety; thus, it does not suppress micronutrient intake from other food sources. The aim of this trial was to evaluate the effects of marine protein hydrolysate (MPH) supplements on physical function and strength in the elderly. This is one of the first long-term studies of MPH and age-related changes in muscle health [12]. 

Our hypothesis was that a daily intake of 3 g MPH for 6 to 12 months would prevent loss of physical performance, compared with a placebo, as measured by the Short Physical Performance Battery (SPPB) and loss of muscle strength as measured by grip strength.

## 2. Results

The mean age of the participants in this study was 72.7 (SD 8.2) years: 72.5 (SD 8.3) and 73.1 (SD 8.2) years in women and men, respectively. Forty participants (46.5%) had higher education, and 30 participants (34.9%) lived alone. Forty-two participants (48.8%) reported that they performed strength exercises at least once a week, and 45 participants (52.3%) reported being physically active daily. 

The mean protein intake was 71.8 g (SD 23.6) and 83.7 g (SD 19.0), corresponding to 1.1 g/kg BW (SD 0.4) and 1.0 g/kg BW (SD 0.2) in women and men, respectively. Energy intake was 1728 kcal (SD 542) in women and 1958 kcal (SD 498) in men.

The mean baseline values of SPPB were 10.5 points (SD 2.4): 10.3 points (SD 2.6) and 11.0 points (SD 1.8) in women and men, respectively. The top SPPB score of 12 points was reached by 44 participants (51.2%). The mean baseline value of grip strength was 33.0 kg (SD 11.6): 26.7 kg (SD 6.7) and 45.5 kg (SD 8.4) in women and men, respectively. Nine women (15.8%) had a grip strength <20 kg, and two men (6.9%) had a grip strength <30 kg. The mean baseline gait speed was 1.0m/s (SD 0.3) in both women and men.

Characteristics of the participants in the intervention group and control group are shown in Table 1, including intake of energy, protein, and seafood, levels of 25-hydroxy vitamin D, outcome measures of physical function, and compliance with intervention/control.

At the 6-month follow-up, 26 participants reported gastrointestinal problems related to the tablets, in the form of increased gas or slight nausea, constipation or diarrhea. Further, 26 participants reported some difficulties swallowing the tablets. However, these problems were apparent in both the intervention and control groups (gastrointestinal symptoms *p* = 0.64 and swallowing difficulties *p* = 0.11), and mean compliance of tablet intake from the start to 6 months was 83.4% (SD 23.3). No compliance differences were found between intervention group and control group (*p* = 0.97).

During the study, nine participants were lost before the 6-month follow-up, and an additional 29 were lost between 6 and 12 months. The participants dropped out for the following reasons: gastrointestinal problems related to tablets (n = 9), health issues not related to the study (n = 4), lack of motivation due to the large number of tablets, no experience of effect, or just ‘had enough’ (n = 10). Three participants developed illnesses listed as exclusion criteria during the study. As illustrated in Table 2, dropouts were older (*p* = 0.01), had lower physical function (SPPB *p* = 0.01 and gait speed *p*< 0.001) and strength (*p* = 0.01). Additionally, dropouts more often lived alone compared with participants completing the study (*p* = 0.03). However, there were no differences in energy, protein, or seafood intake between dropouts and participants completing the study. 

According to the primary analysis based on independent samples *t*-test, there was no significant difference in mean change between intervention and control groups in the main outcome, SPPB, from baseline to 12 months (*p* = 0.41). However, this analysis does not adjust for differences in the cluster effect within each municipality, which was found to be of significance with ICC 88.7%, 37.0%, and 62.3% for SPPB, grip strength, and gait speed, respectively. Particularly, the participants recruited from homecare services were very distinct from the others.

According to a bivariate linear mixed model for SPPB adjusting for a cluster effect on a municipal level, there was no overall difference between the intervention group and control group regarding time trend in SPPB, grip strength, or gait speed (non-significant interactions) (Table 3). The intervention and control groups were, however, significantly different in SPPB at the baseline (*p* = 0.033) and 6-month follow-up (*p* = 0.037), but not at the 12-month follow-up (*p* = 0.093). For grip strength, groups were not significantly different at any time point. Regarding gait speed, groups were significantly different at the baseline (*p* = 0.027) and 6-month follow-up (*p* = 0.027), but not at the 12-month follow-up (*p* = 0.106). These results are illustrated with 95% confidence intervals in Figure 1.

Due to dropout during the follow-up period, a number of preplanned covariates had to be reconsidered. Protein and energy intake were excluded from the model as they highly correlated with each other (r = 0.75) and were not significantly associated with the outcome variables at the baseline [13]. Vitamin D measurements were performed with 6 months between analyses to account for seasonal variation; however, 21 participants were missing one of the two measurements. Seven participants were missing both measurements of vitamin D. Thus, vitamin D was also excluded as an adjustment covariate. The results of the multiple linear mixed model reduced by Akaike’s Information Criterion did not change the conclusions of the bivariate analyses.

## 3. Discussion

Our intervention study among elderly persons did not reveal any significant differences in measures of physical function (SPPB and gait speed) or strength (grip strength) between the intervention group receiving a supplement of MPH and the control group receiving a placebo. However, the study participants had good physical function and strength and sufficient protein intake at the starting point. We used standardized measurements and questions to collect data at several time points, ensuring comparability of results across different studies.

Mean SPPB in this study was 10.3 points in women and 11.0 points in men. In the power calculation, we anticipated the mean SPPB to be 7.5 (see methods Section 4.7). Our assumption was based on data from a more sedentary and already frail population [14] as the initial goal was to recruit participants among home care service users. The SPPB scores observed in our study were more in line with the normative scores reported by Bergland et al. [15], where the mean score in the age group of 70–74 years was 10.8 in women and 11.4 in men. Bergland’s study also reports that the SPPB test has a considerable ceiling effect, where more than 20% have the highest or lowest scores. In our study, 51.2% of the participants reached the top score of 12 points. Thus, the ceiling effect of the SPPB test is a considerable shortcoming of our study as possible changes in physical function could not be identified by our primary outcome measure.

The baseline values of the secondary outcomes, grip strength and gait speed, further underline that the participants in our study sample have good physical function and strength. The mean grip strength in our study was 26.7 kg in women and 45.5 kg in men. Compared with the normative data reported by Dodds et al., this represents a level of grip strength comparable to 60-year-old women and men [16]. However, all participants in our study were >65years, with a mean age of 72.7 years. Recent studies in Norway [17] and Finland [18] show that older adults are stronger and have better grip strength now than in earlier generations, corresponding to a five-year difference, i.e., the more recently born generation of 80-year-olds have a similar mean grip strength as 75-year-olds born one generation earlier [17]. 

The participants in our study had a protein intake of 1.1 g/kg BW, which is higher than the recommended daily allowance (RDA) of 0.8 g/kg and in accordance with the increased dosage for elderly persons according to Nordic nutritional recommendations [19]. Protein intake is, however, lower than in other Norwegian studies such as Norkost3 [20] and the Tromsø study [21]. Energy intake in our study was in line with the results of Norkost3 in women; however, the male participants in our study reported lower energy intake compared with Norkost3. This might indicate underreporting, especially by men. Protein and energy intake were, however, assessed by a single day food recall, and this is a weakness as diet often varies from day to day. Protein and energy intake were not included in the regression model. However, their sufficient protein intake might indicate that this study population was not the appropriate target group for MPH supplementation.

Moreover, the participants in our study had a relatively high seafood intake. The mean computed seafood index corresponded to >3 meals of seafood per week, which is in accordance with the dietary guidelines in Norway [22] and in line with data from Norkost3 [20]. The seafood intake was equally high in both intervention and control groups, and thus would not influence the results in terms of difference between groups. However, the relatively high seafood intake and protein intake among the participants might indicate that they were not in need of supplementation, and thus it was difficult to notice any effects.

We anticipated that the SPPB score would decline by 0.9 points in the control group. However, we could not detect a change in physical function over time in either the control or intervention group. Bergland et al. [15] demonstrated that a decline in physical function, as measured by the SPPB, occurs in the mid-sixties, with a slightly earlier decline in women than in men. However, Hämäläinen [23] examined 6-year changes in physical performance among high-functioning older adults and found improvement in physical performance in age groups comparable to the participants in our study. Hämäläinen finds that this might be related to an increased physical activity level after recent retirement from work. In our study, more than 50% reported being physically active daily. Weekly strength exercise was also frequently reported. However, only self-reported frequency of physical activity and strength exercise was measured, not the loading or the level of activity. Strength exercise might be the most influential factor in muscle health in ageing, and nutritional supplements may support these effects [24]. We wanted to examine the effects of the MPH supplement without exercise; however, the possible high frequency of strength exercise in the group might have biased the results. 

Daily physical activity was more frequent in the intervention group than in the control group, and the intervention and control groups were also significantly different in SPPB and gait speed at the baseline and the 6-month follow-up, with higher scores in the intervention group according to linear mixed model analyses adjusting for a cluster effect on a municipal level. Groups were not significantly different at the 12-month follow-up. Thus, differences between the intervention and control groups decreased with time; however, this could be related to selection bias as dropouts were significantly different from those completing the full 12-month follow-up. 

Participants dropping out of the study had significantly lower scores in SPPB, grip strength, and gait speed compared with those completing the follow-up. A review of attrition in longitudinal studies among the elderly shows that dropout is associated with, e.g., higher age, fewer years of education, poor functioning, and living alone [25]. These factors are also significantly different between participants dropping out versus those completing our study. However, dietary factors did not significantly differ between dropouts and those completing the study, nor did compliance in intervention or problems related to the tablets. Compliance was high in both the intervention group and control group, despite many reported problems with swallowing and/or gastrointestinal effects related to the tablets. One out of three participants experienced gastrointestinal problems, and every fourth participant had difficulties swallowing the tablets. As there were no differences in gastrointestinal complaints or swallowing difficulties between the intervention group and control group, we believe that the problems might be related to the large number of tablets and possibly to the additives used to make the tablets look similar and be odor free. 

This study did not identify differences in measures of physical performance and strength between the intervention group and control group; however, this might be related to the limitations of the study. As previously discussed, the study participants had good physical function, strength, and nutrient intake, and we could not see a decline in physical function during the study. This might indicate that they were not in need of a supplement enhancing MPS. On the other hand, one year might not be sufficient time to detect the preventive effect on muscle health. This was one of the first studies assessing MPH in relation to physical function and strength in older adults [12], and we suggest the following recommendations for future studies: -Future studies should be performed on a frailer population or on populations with immobilized older adults. The population that is most in need of help with stimulating MPS is composed of people who are immobilized over a period of time, e.g., after injuries or illnesses affecting mobility.-MPH might have the potential to mitigate loss of muscle function and strength in populations of older adults with lower protein and seafood intake.-Physical activity should be measured more precisely.-The large number of tablets was burdensome and might have caused gastrointestinal symptoms in both the intervention and control groups. The number of tablets should be reduced. The use of a soft drink, similar to the supplement used in the feasibility study by Drotningsvik et al. [26], might be a better choice. Alternatively, MPH could be used as an additive in enteral nutrition solutions or soft drink supplements.

## 4. Materials and Methods 

This study was a randomized, controlled, double-blinded trial in which the participants received tablets containing 3 g of MPH or a placebo, with data collected at a baseline and at 6- and 12-month follow-ups. A 12-month follow-up period was planned for all participants; however, they were given the supplements for 6 months at a time, with the opportunity to withdraw at any time. Thus, statistical analyses were pre-planned to include measurements at both 6 months and 12 months of intervention. The study design is graphically illustrated in Figure 2.

### 4.1. Recruitment

Participants were recruited from March 2017 to May 2018, with all follow-up observations completed in January 2019. The study included adults ≥65 years old. Exclusion criteria included active cancer or progressive muscle illness (e.g., multiple sclerosis or Lou Gehrig’s disease), diabetes, kidney failure, short life expectancy (<1 year), and allergies to fish protein. Mental illness or neurodegenerative illnesses were not defined as exclusion criteria, however; participants were considered competent enough to provide informed consent to participate. Initially, one of the inclusion criteria was need of home care services, and recruitment was supposed to be facilitated trough municipal healthcare services. However, this recruitment proved difficult, and after 6 months and only 16 participants recruited, we omitted the criterion of the need for municipal support and started recruitment through local media and by leaving flyers at healthcare offices and senior citizen associations. A total of 92 individuals from several municipalities on the west coast of Norway consented to participate in the study. They represented a mixed cohort of elderly receiving home care services (n = 16) and elderly who were independent in daily activities and thus were without need of municipal home care (n = 76). The predetermined sample size goal of 82 (see power calculation below) was achieved, however, we did not succeed in recruiting extra participants to account for dropouts. The flow of participant recruitment and allocation is described in Figure 3 (CONSORT flow diagram).

### 4.2. Randomisation

Eight participants withdrew their consent, and one was excluded due to exclusion criteria occurring after inclusion, leaving 86 participants, 57 women and 29 men for randomization into the intervention group (n = 43) or control group (n = 43). 

The participants were given project identification numbers and block-randomized into two groups. Blocks were based on the baseline score of the SPPB test (low score 0–6 points, medium score 7–9 points, or high score 10–12 points) and gender. The main researcher (LKN) enrolled participants and forwarded identification numbers and block information to the statistician (JSB), who performed randomization into either group A (n = 43) or group B (n = 43). Supplements and placebos were distributed in similar boxes prepared by a person not involved in the study and only marked with participants’ identification numbers. Participants, care providers, outcome assessors, and data analysts were blinded for the intervention. The study investigators were blinded when analyzing the primary and secondary outcomes as the code was masked until statistical analyses were completed.

### 4.3. Intervention

The participants in both the intervention and control groups were instructed to take five tablets twice a day, preferably with breakfast and their evening meal. For the intervention group, each tablet contained 300 mg of MPH, which corresponds to a total dose of 3 g of protein per day (1.5 g in the morning and 1.5 g in the evening). The tablets were produced by Flexipharma AS and based on the marine peptide compound 565952 P from Firmenich Bjørge Biomarin AS (data sheet in Appendix A). The marine peptide compound was manufactured through the hydrolysis of fresh or fresh-frozen Atlantic cod fillet (*Gadus morhua*) using industrial food approved non-GMO proteolytic enzymes, and maltodextrin from corn was added to mask the odor and taste of fish. Hydrolysis was conducted using equipment and procedures according to regulations provided by the Norwegian Food Safety Authorities. Marine peptides are approved as a food ingredient in Norway according to EU regulations. The control group received the same number of similar-looking placebo tablets produced from gum Arabic.

### 4.4. Assessments

Assessments were completed in the participants’ homes at the baseline and at 6- and 12-month follow-ups. The researcher filled in the questionnaires based on a structured interview with each participant. Demographic data included age, gender, living condition, and educational level, and this information was collected only at the baseline. The participants’ education levels were dichotomized as having higher or lower education. Higher education was defined as education at university or college level, i.e., more than 12 years of education. 

Physical activity was reported in categories based on how often participants performed strength exercises and how often they were physically active otherwise (e.g., walking, running, bicycling). Strength exercise was dichotomized into at least once a week or more seldom. Other physical activity was dichotomized into daily or more seldom.

A 24-hour food recall was conducted using a multiple pass method following the five-step protocol described by Moshfegh et al. [27], recording all food items and amounts eaten the day before the visit. To adapt the method to Norwegian participants, we used an illustrated food portion booklet with a corresponding list of weights from the Norwegian study Ungkost 2000 [28]. Energy and protein intake per day were calculated using the online diet tool from the Norwegian Directorate of Health and the Norwegian Food Safety Authority, www.kostholdsplanleggeren.no. The food database in the Diet Planner is based on the Norwegian Food Composition table, which provides an overview of the content of energy and nutrients for the most common foods eaten in Norway.

Estimated protein and energy intake were expressed in grams per kilogram (g/kg) of body weight (BW). For participants who were under- or overweight (BMI < 22.0 kg/m^2^ or >27 kg/m^2^), the BW was adjusted by applying the BW corresponding to a BMI of 22 or 27 kg/m^2^, respectively. Higher BMI is recommended for older adults compared with adults [29,30]. The BMI cut-off values used in this study are described in the Norwegian guidelines for treatment and prevention of malnutrition [31]. This adjustment was made to let the intake represent the intake related to protein requirement rather than BW in under- and overweight individuals, as underweight persons require extra protein to build muscle tissue, while the extra weight in overweight persons is often composed of adipose tissue [32,33].

Habitual intake of seafood in the past 6 months was estimated by using a shorter version of a previously validated Food Frequency Questionnaire (FFQ) [34]. The frequency of seafood intake for dinner and frequency of seafood used as a spread in salads, as a snack, or something similar was recorded. Responses were reported as never, <1 time/month, 1–3 times/month, once/week, 2–3 times/week, and ≥4 times/week. The midpoint of categories was used to calculate frequency of meals per week, e.g., 2–3 times per week was regarded as 2.5. Frequency of seafood used as a spread in salads, as a snack, or something similar was divided by six, as six portions of seafood as spreads correspond to one dinner portion. Thus, the combined frequency of seafood consumption corresponds to a dinner portion equivalent per week. This method of making a continuous scale of seafood consumption from FFQ was developed and validated against biomarkers by Markhus et al. [35].

Body mass was recorded in light clothing to the nearest 0.1 kg using a digital scale (Seca 803), and height was measured with a tape measure (Seca 201). Furthermore, the participants were instructed to visit their general practitioner (GP) for blood sampling. Blood samples were handled at the participants’ local municipality laboratories, and serum was sent to the local hospital for analysis of 25-hydroxy vitamin D as a biomarker for vitamin D status. 

### 4.5. Main Outcome: Short Physical Performance Test (SPPB)

Muscle performance was measured with the SPPB, a test battery developed originally for the Established Populations for Epidemiologic Studies of the Elderly and translated into Norwegian in 2013 by Bergh and colleagues [36]. The test battery includes standing balance, walking speed, and repeated chair rise. Each of the three domain scores ranges from 0–4 points, yielding an integer sum score ranging from 0–12 points. A higher sum score indicates a higher level of functioning, and a change of one unit is considered a clinically meaningful change [14]. A systematic review of instruments assessing performance-based physical function in older community-dwelling persons concluded that the SPPB is highly recommended in terms of validity, reliability, and responsiveness [37]. The SPPB complements self-reported disability and may predict mortality and nursing home admission even at the high end of the functional spectrum [38]. Gait speed was calculated as meters per second (m/s) from the 4m walking test included in SPPB and used as a secondary outcome. 

### 4.6. Secondary Outcome: Grip Strength

Grip strength was measured to the nearest 0.1 kg using the Jamar Plus+ digital hand dynamometer. The maximum value of three trials on each hand was used for analyses [16]. Grip strength is a useful and simple measure of muscle strength. It correlates with leg strength, is a clinical marker of poor mobility, and is considered a better predictor of clinical outcomes than low muscle mass [39]. Cutoff scores are set at 30 kg and 20 kg for men and women, respectively [39]. However, Dodds et al. provided normative data for grip strength across the life course [16]. 

### 4.7. Sample Size and Statistical Power Calculation 

Based on an American study reporting a clinically meaningful change in the main outcome of SPPB to be 0.4–1.5 points [14], we assumed that the mean SPPB score at inclusion was 7.52 in both groups, and the standard deviation after the intervention period would be 1.42 in both groups. Freiberger et.al. [37] evaluated the responsiveness of SPPB, and the observed effect sizes in intervention studies were ranging from 0.48 to 1.25. Based on these numbers, we assumed a reduction of 0.9 (mid-point) in the control group. We also assumed that the intervention group maintained a stable SPPB score throughout the study period, so that the effect would be seen as a decrease in the control group. A sample size of 41 participants in each group was required to detect a statistically significant mean difference of 0.9 points at a significance level of 5% and with a power of 80% using a two-sided independent samples *t*-test. As the participants were older, we expected a 20% dropout rate and therefore aimed to include 50 participants in each group. 

### 4.8. Statistical Analyses

The demographic and clinical characteristics of the participants in each group were described as means and standard deviations (SDs) or frequencies and percentages, as appropriate. Independent samples *t*-test and χ^2^-test were used to assess the gender differences and differences between participants completing the study and the dropouts. As a primary analysis, independent samples *t*-test was used to assess the difference in change in the primary outcome, SPPB, between the intervention group and the control group.

A linear mixed model was then estimated to assess the effect of MPH on the SPPB in the intervention group and control group throughout the 12-month follow-up. The model included fixed effects for time components up to the second order to account for non-linear effects, group allocation, and interaction between the two. A significant interaction between time and group allocation would imply differences in SPPB scores between the groups throughout the follow-up. As participants were recruited from different municipalities, the data will likely exhibit a hierarchical structure. The within-municipality cluster effect was assessed using the intra-class correlation coefficient (ICC). Moreover, due to repeated measurements for each participant, the within-participant correlation will likely be present. Therefore, random effects for participants nested within the municipality were included to correctly adjust estimates for possible within-participant and within-municipality correlations. Random slopes for the time component were considered but not included as they did not improve the model fit. Pairwise comparisons were conducted by deriving individual time-point contrasts within each group with corresponding 95% confidence intervals (CIs) and *p*-values, and the results were illustrated graphically. The results were further adjusted for several of the pre-planned covariates: age, gender, education, living status, strength exercise, and habitual seafood intake [40]. Some of the pre-planned covariates were not included due to high correlations or many missing values. Cases with missing values on one or more covariates were excluded from regression analysis. Akaike’s Information Criterion, where the smaller value means a better model, was used to reduce models for excessive adjustment variables. The secondary outcomes grip strength (kg) and gait speed (m/s) were assessed in the same way. Statistical analyses were performed using SPSS v 24 and SAS v 9.4. Results with *p*-values below 0.05 were considered statistically significant. The intention-to-treat principle was used in statistical analysis.

### 4.9. Ethical Approval and Registration 

All subjects gave their informed oral and written consent for inclusion before they participated in the study, which was conducted in accordance with the Declaration of Helsinki and approved by the Regional Committee on Ethics in Medical Research (REK) in Mid-Norway in September 2016 with the registration ID 2016/1152. Changes in inclusion criteria were approved in August 2017. The study protocol has been published elsewhere [40], and the trial was pre-registered at http://www.clinicaltrials.gov with the unique identifier NCT02890290.

## Figures and Tables

**Figure 1 marinedrugs-19-00062-f001:**
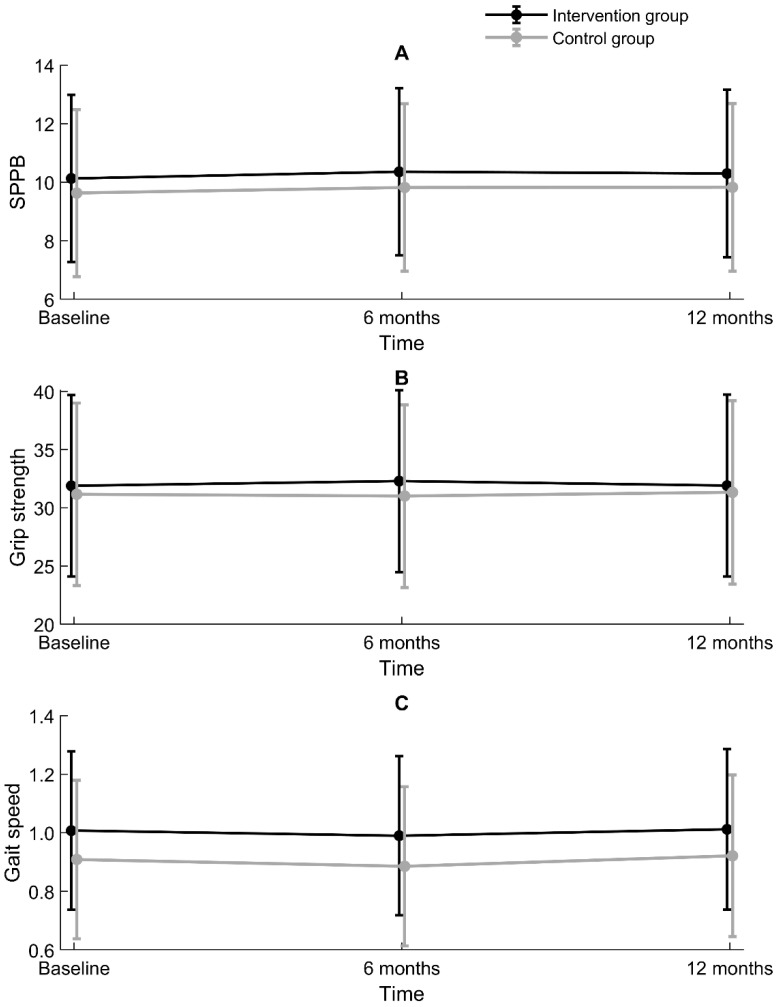
Differences between the intervention group and control group for (**A**) Short Physical Performance Battery (SPPB, points), (**B**) grip strength (kg), and (**C**) gait speed (m/s), with a 95% confidence interval, estimated by linear mixed models adjusting for within-participant and within-municipality correlations.

**Figure 2 marinedrugs-19-00062-f002:**
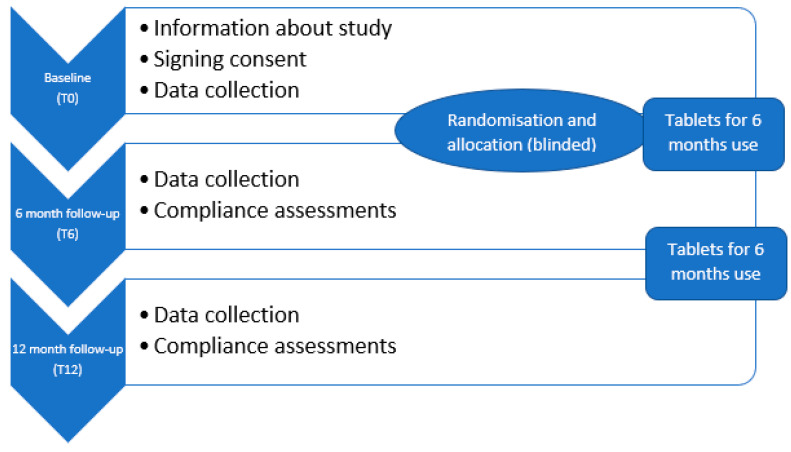
Study design: A randomized, controlled, double-blinded trial with data collection at the baseline, 6 months and 12 months.

**Figure 3 marinedrugs-19-00062-f003:**
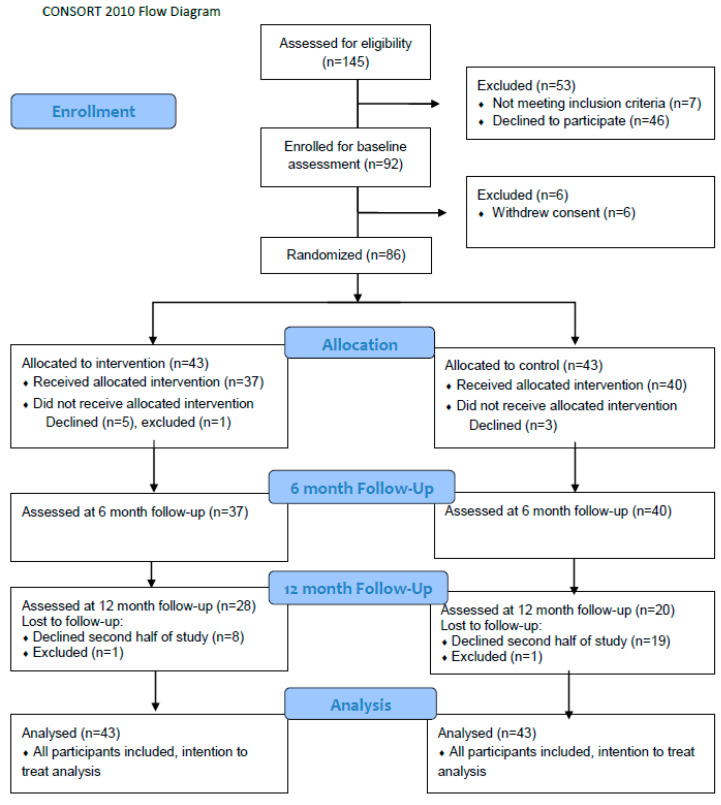
Consort flow diagram illustrating flow of participant recruitment and allocation.

**Table 1 marinedrugs-19-00062-t001:** Demographic and clinical characteristics at the baseline, and descriptive statistics of the outcome variables of the randomized participants (N=86) at each time point, in the intervention (n = 43) * and control (n = 43) * groups.

	Intervention	Control
Demographic and clinical characteristics		
Demographic: Age in years, mean (SD)	73.4 (8.7)	72.0 (7.7)
Weight, kg mean (SD)	72.2 (13.1)	78.5 (14.7)
BMI, kg/m^2^ mean (SD)	25.5 (3.8)	27.7 (4.7)
Gender, female, n (%)	29 (67.4)	28 (65.1)
Education level college/university, n (%)	21 (48.8)	19 (44.2)
Perform strength exercise weekly, n (%)	22 (51.2)	20 (46.5)
Other physical activities daily, n (%)	28 (65.1)	17 (39.5)
Live alone, n (%)	16 (37.2)	14 (32.6)
Dietary factors		
Energy intake, kcal, mean (SD)	1809 (551)	1802 (527)
Energy intake, kcal/kgBW ^1^, mean (SD)	26.4 (8.8)	25.2 (8.8) ^a^
Protein intake, g/kgBW ^1^, mean (SD)	1.1 (0.3)	1.1 (0.4) ^a^
Protein intake, g/1000 kcal, mean (SD)	42.3 (10.4)	43.5 (10.1)
Seafood index ^2^, mean (SD)	3.3 (1.4) ^a^	3.4 (1.2)
Serum 25-hydroxy vitamin D, nmol/L, mean (SD)	84.2 (31.3) ^c^	84.1 (31.8) ^b^
Outcome variables		
SPPB ^3^, total points, mean (SD)		
baseline	10.7(2.3)	10.4 (2.4)
6 months	11.0 (2.1) ^d^	11.0 (1.6) ^c^
12 months	11.4 (1.8) ^f^	11.1 (1.8) ^g^
Grip strength, kg, mean (SD)		
baseline	33.0 (11.4)	33.1 (11.9)
6 months	34.2 (12.6) ^d^	34.1 (10.6) ^b^
12 months	37.0 (11.5) ^f^	33.9 (10.9) ^g^
Gait speed, m/s, mean (SD)		
baseline	1.1 (0.3)	1.0 (0.3)
6 months	1.0 (0.3) ^d^	1.0 (0.2) ^c^
12 months	1.1 (0.2) ^f^	1.1 (0.2) ^g^
Compliance % of tablets taken, 0–6 months (SD)	83.5 (23.1)	83.4 (23.8)
% of tablets taken, 6–12 months (SD)	92.1 (18.4)	92.9 (5.8)
Difficulties swallowing tablets n (%)	16 (29.7) ^d^	10 (25.6) ^c^
Gastrointestinal effects of tablets n (%)	11 (32.4) ^e^	15 (37.5) ^b^

^1^ BW = body weight, adjusted for over- and underweight. ^2^ Seafood index is computed from frequency questions regarding seafood for dinner and lunch, and it corresponds to dinner portion equivalents per week. ^3^ Short Physical Performance Battery. * Variables with missing values are marked with remaining n: ^a^ n = 42, ^b^ n = 40, ^c^ n = 39, ^d^ n = 37, ^e^ n = 34, ^f^ n = 28, ^g^ n = 20.

**Table 2 marinedrugs-19-00062-t002:** Baseline characteristics of, and *p*-value for difference between, participants completing the 12-month follow-up (n = 48) and participants dropping out (n = 38).

	Completing	Drop-Out	*p*-Value	
Age in years, (mean ± SD)	70.5 (6.3)	75.5 (9.5)	0.01 ^2^	
Female gender, n (%)	28 (58.3)	29 (76.3)	0.08 ^3^	
Living alone, n (%)	12 (25.0)	18 (47.4)	0.03 ^3^	
Education level college/university, n (%)	27 (56.3)	13 (34.2)	0.04 ^3^	
Intervention group, n (%)	28 (58.3)	15 (39.5)	0.08 ^3^	
Control group, n (%)	20 (41.7)	23 (60.5)	0.08 ^3^	
Nutrition				
Protein intake, g/kgBW ^1^	1.1 (0.4)	1.1 (0.3)	0.98 ^2^	
Energy intake, kcal/kgBW ^1^	25.9 (10.2)	25.8 (6.6)	0.98 ^2^	
Seafood intake, index	3.5 (1.3)	3.2 (1.2)	0.38 ^2^	
Physical function				
SPPB, points (mean ± SD)	11.2 (1.5)	9.7 (3.0)	0.01 ^2^	
Grip strength, kg (mean ± SD)	35.7 (11.2)	29.6 (11.3)	0.01 ^2^	
Gait speed, m/s (mean ± SD)	1.1 (0.3)	0.9 (0.3)	<0.001 ^2^	

^1^ BW = body weight, adjusted for over- and underweight. ^2^
*p*-value for independent samples *t*-test. ^3^
*p*-value for χ^2^-test.

**Table 3 marinedrugs-19-00062-t003:** Results of linear mixed models assessing changes in Short Physical Performance Battery (SPPB), grip strength, and gait speed in relation to the intervention or control group. Adjusted for a cluster effect on a municipal level.

	SPPB				Grip Strength				Gait Speed			
	N = 196 (n = 85 at T0, n = 68 at T6, n = 43 at T12)				N = 197 (n = 85 at T0, n = 68 at T6, n = 43 at T12)				N = 196 (n = 85 ved T0, n = 68 ved T6, n = 43 ved T12)			
	Bivariate Models		Multiple Model		Bivariate Models		Multiple Model		Bivariate Models		Multiple Model	
	Regr.coeff (SE)	*p*-Value	Regr.coeff (SE)	*p*-Value	Regr.coeff (SE)	*p*-Value	Regr.coeff (SE)	*p*-Value	Regr.coeff	*p*-Value	Regr.coeff (SE)	*p*-Value
Intercept Time	10.1 (1.5)	0.01	11.08 (2.4)	<0.001	31.9 (4.0)	0.003	65.6 (12.2)	<0.001	1.0 (0.1)	0.01	2.4 (0.4)	
T0	0		0		0		0		0		0	
T6	0.2 (0.2)	0.25	0.2 (0.2)	0.23	0.4 (0.6)	0.5	0.5 (0.6)	0.45	−0.02 (0.003)	0.57	−0.01 (0.003)	0.80
T12	0.2 (0.2)	0.30	0.1 (0.2)	0.42	0.01 (0.5)	0.98	0.2 (0.6)	0.76	−0.0052 (0.04)	0.90	−0.0002 (0.04)	0.99
Group												
Intervention group–ref	0		0		0		0		0		0	
Control group	−0.5 (0.2)	0.030	−0.5 (0.2)	0.030	−0.7 (2.2)	0.75	−1.2 (−1.2)	0.31	−0.1 (0.04)	0.03	−0.1 (0.04)	0.02
Time x Group												
T0	0		0		0		0		0		0	
T6	−0.04 (0.3)	0.89	−0.1 (0.3)	0.80	−0.6 (0.8)	0.50	0.5 (0.8)	0.51	−0.01 (0.04)	0.88	−0.02 (0.04)	0.65
T12	0.03 (0.2)	0.99	0.1 (0.2)	0.83	0.2 (0.8)	0.85	0.04 (0.8)	0.96	0.01 (0.1)	0.89	−0.004 (0.1)	0.94
Age	−0.02 (0.02)	0.46	−0.02 (0.03)	0.39	−0.1 (0.3)	0.74	−0.3 (0.2)	0.07	−0.02 (0.01)	0.003	−0.02 (0.01)	0.01
Gender, female	−0.2 (0.2)	0.49	−0.2 (0.2)	0.33	−17.8 (1.2)	<0.001	−018.3 (1.3)	<0.001	0.02 (0.04)	0.61	−0.03 (0.04)	0.46
Education, higher	0.5 (0.02)	0.02	0.6 (0.2)	0.01	−1.0 (2.5)	0.70	0.4 (1.3)	0.76	0.1 (0.04)	0.07	0.1 (0.04)	0.08
Strength exercise	0.1 (0.2)	0.40	0.1 (0.2)	0.39	−4.6 (0.6)	0.51	−0.2 (0.5)	0.71	0.1 (0.03)	0.05	0.1 (0.03)	0.02
Live Alone	−0.1 (0.2)	0.58	−0.1 (0.3)	0.85	−7.2 (2.3)	0.003	−0.2 (1.5)	0.91	−0.03 (0.04)	0.46	0.02 (0.1)	0.61
Seafood intake	−0.1 (0.1)	0.33	−0.1 (0.1)	0.40	0.3 (0.3)	0.23	0.4 (0.3)	0.09	0.01 (0.01)	0.68	0.01 (0.1)	0.57

## Data Availability

The data presented in this study are available on request from the corresponding author. The data are not publicly available due to privacy restrictions.

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
