# Peer review of "Limited Benefit of Marine Protein Hydrolysate on Physical Function and Strength in Older Adults: A Randomized Controlled Trial"

_marinedrugs, 2021, doi:10.3390/md19020062_

Round 1

Reviewer 1 Report

This study was performed to evaluate the effects of marine protein hydrolysate (MPH) supplement on physical function and strength in the elderly by using the double-blinded, randomised, controlled trial. The trial was carried out properly and the results were presented precisely. Unfortunately, the study could not identify differences in measures of physical performance and strength between intervention group and control group. The authors pointed out several reasons, concerning with study participants, sample shape, etc.  As they discussed, future studies should be designed properly on the basis of their findings obtained in this study. 

Reviewer 2 Report

The paper by Nygard et al. describes a randomised control intervention study, in which the authors tested the hypothesis that marine protein hydrolysate (MPH) would prevent the loss of physical function over 6-12 months in elderly men an women.

The manuscript is well written for the most part, with only minor language edits required and the authors should be commended on undertaking a longitudinal intervention study such as this. Overall the data is sound and the study design is appropriate, but there are some issues/limitations that should be at least commented upon by the authors.

  1. The authors chose to supplement with only 3 g of MPH per day, but the rationale for this is weak in my opinion. In the introduction there is some discussion about the ability of 3-4 g of leucine to protect from muscle loss during immobilisation but in fact, the paper referenced used 3-4g (more like 4 g if you do the maths) per meal. This would equate to approximately 12 -13 g of extra leucine per day. 
  2. The referenced used to make the point above (reference 8) is a review paper. It is more appropriate to cite the original article by English et al. 2016 Am J Clin Nutr. However given the issues highlighted above, the authors may want to reconsider the inclusion of this paper as a rationale for the dose used.
  3.  On the same point, the authors should give detail (maybe in a table) of the amino acid composition of the MPH used. Since the rationale is at least in part derived on the fact that MPH is a good source of leucine, we need to know how much leucine it has in it.
  4. In addition the others cite the paper by Vikoren et al. 2013 (reference 9)- and in this study they used 3 g initially and then 6 g for the final 4 weeks. 
  5. Hence overall, i feel that some discussion is warranted around the dose provided. In my opinion i think it was too low to see any effect.
  6. Clearly a major draw back to the authors ability to truly scrutinise the effects of the supplement is the fact that the placebo group suffered no decline in physical performance. I am intrigued to know why the authors believed there would be an average reduction in SPPB score of 0.9? This assumption does not appear to be justified anywhere.
  7. it is a shame, given the immense effort that this study must have taken, that there are no additional measures of muscle. in particular some crude measures (or ideally more advanced measures) of muscle mass/size would have been a great addition. Perhaps the authors could comment on why this was not included. This may also aid there discussion somewhat since it is well established that there is often a discrepancy between muscle mass and muscle function/physical function.

Reviewer 3 Report

Review of the manuscript (manuscript ID: marinedrugs-1034794) titled: Limited benefit of marine protein hydrolysate on physical function and strength in older adults: a randomized controlled trial

Thank you for the opportunity of review the manuscript of Nygård et al. who carried out the RCT to determine the effects of marine protein hydrolysate (MPH) on physical function in older adults. While the study is interesting, but a serious methodological problems have existed and the manuscript needs many changes.

Specific critical comments are presented below:

- What was the basis of choosing such a dose of MPH (3 g/day)? Could 3 g of MPH be physiologically relevant dose to prevent sarcopenia? The Authors need to explain it in the text. The same suggestion regards the period of intervention. Why such a long period of intervention was applied?

- Reading the text I am not sure how long was the intervention: 6 or 12 months? Was the follow-up period equal for all the participants?

- Material and Methods:

  1. The recruitment process needs to be describe more detailed (information about cohort, or the region of origin of participants…)
  2. Were advanced mental illness or neurodegenerative disorders considered as an exclusion criterion?
  3. Recruitment and randomization procedure should be described as a separate subsections;
  4. Graphical study design is highly recommended;
  5. Was the body composition (BIA) of the participant assessed? This additional parameter would significantly increase the quality of the research.
  6. What method was used production for protein hydrolysate. This should be explained in the text.
  7. The amino acid composition of marine protein hydrolysate is needed to be added to the manuscript

- Results:

  1. First two sentences (lines 217-219) should be moved to section 2.1;
  1. The subjective side GI effects of taking MPH reported by participants of intervention and placebo group should be clearly described in the text;
  2. The results require huge improvement and arrangement. The clinical characteristic of participants (Physical function) presented in table 1 are repeated at the Figure 2
  3. Table 1 should present a baseline characteristic of participants with statistical analysis, a the next ones or Figures should reflect the results of intervention;
  4. The protein intake should be also calculated as g/1,000 kcal;
  5. The results of intervention versus placebo groups must be improved and should present as time point analysis (0 month vs. after 6 months of follow-up, and after 6 vs. 12 months of follow up)
  6. What was the drop-out rate in intervention group?
  7. Table 3: I suppose that some terms are in Norwegian;

- Discussion:

  1. The section must be re-written taking account a re-written results.
  2. How the relatively high seafood intake could influence on obtained results of the study?
  3. Lines 326-327: the sentence is speculative. It is difficult to assess the leucine intake without the data of leucine content in the tested hydrolysate.

- Study limitations and Conclusions need to be clearly distinguished.

- Conclusion and abstract should be improved after correction the results.

Round 2

Reviewer 3 Report

The Authors properly addressed all my comments and suggestions. I accept the manuscript for publication.